

# Rare ground data confirm significant warming and drying in western equatorial Africa

Emma R. Bush[1], Kathryn Jeffery[1,2], Nils Bunnefeld[1], Caroline Tutin[1],
Ruth Musgrave[3], Ghislain Moussavou[4], Vianet Mihindou[2,5],
Yadvinder Malhi[6], David Lehmann[1,2], Josué Edzang Ndong[2],
Loïc Makaga[2] and Katharine Abernethy[1,7]

[1] Faculty of Natural Sciences, University of Stirling, Stirling, UK
[2] Agence Nationale des Parcs Nationaux (ANPN), Libreville, Gabon
[3] Elephant Protection Initiative, London, UK
[4] Agence Gabonaise d'Études et d'Observation Spatiale (AGEOS), Libreville, Gabon
[5] Ministère des Eaux et Forêts, Charge de l'Environnement et du Développement Durable, Libreville, Gabon
[6] Environmental Change Institute, School of Geography and the Environment, University of Oxford, Oxford, UK
[7] Institut de Recherche en Écologie Tropicale, CENAREST, Libreville, Gabon

Corresponding author
Emma R. Bush, e.r.bush@stir.ac.uk

## ABSTRACT

**Background:** The humid tropical forests of Central Africa influence weather worldwide and play a major role in the global carbon cycle. However, they are also an ecological anomaly, with evergreen forests dominating the western equatorial region despite less than 2,000 mm total annual rainfall. Meteorological data for Central Africa are notoriously sparse and incomplete and there are substantial issues with satellite-derived data because of persistent cloudiness and inability to ground-truth estimates. Long-term climate observations are urgently needed to verify regional climate and vegetation models, shed light on the mechanisms that drive climatic variability and assess the viability of evergreen forests under future climate scenarios.
**Methods:** We have the rare opportunity to analyse a 34 year dataset of rainfall and temperature (and shorter periods of absolute humidity, wind speed, solar radiation and aerosol optical depth) from Lopé National Park, a long-term ecological research site in Gabon, western equatorial Africa. We used (generalized) linear mixed models and spectral analyses to assess seasonal and inter-annual variation, long-term trends and oceanic influences on local weather patterns.
**Results:** Lopé's weather is characterised by a cool, light-deficient, long dry season. Long-term climatic means have changed significantly over the last 34 years, with warming occurring at a rate of +0.25 °C per decade (minimum daily temperature) and drying at a rate of −75 mm per decade (total annual rainfall). Inter-annual climatic variability at Lopé is highly influenced by global weather patterns. Sea surface temperatures of the Pacific and Atlantic oceans have strong coherence with Lopé temperature and rainfall on multi-annual scales.
**Conclusions:** The Lopé long-term weather record has not previously been made public and is of high value in such a data poor region. Our results support regional analyses of climatic seasonality, long-term warming and the influences of the oceans on temperature and rainfall variability. However, warming has occurred more

rapidly than the regional products suggest and while there remains much uncertainty in the wider region, rainfall has declined over the last three decades at Lopé. The association between rainfall and the Atlantic cold tongue at Lopé lends some support for the 'dry' models of climate change for the region. In the context of a rapidly warming and drying climate, urgent research is needed into the sensitivity of dry season clouds to ocean temperatures and the viability of humid evergreen forests in this dry region should the clouds disappear.

# INTRODUCTION

The humid forests of Central Africa make up 30% of the world's tropical forests (*Malhi et al., 2013*), are a major carbon store (*Lewis et al., 2013*) and influence weather globally (*Bonan, 2008*; *Washington et al., 2013*). Most African evergreen tropical forests are found in the western equatorial region where total annual rainfall is less than 2,000 mm (*Philippon et al., 2019*). Evergreen forests can be maintained in this relatively dry region due to reduced water demand during seasonal drought associated with extreme cloudiness (*Philippon et al., 2019*). Long-term changes to climate and climatic variability in the region (*James, Washington & Rowell, 2013*) are likely to have far-reaching impacts on the functioning of these evergreen tropical forests (*Asefi-najafabady & Saatchi, 2013*; *Zhou et al., 2014*) with knock-on effects for the global carbon cycle (*Mitchard, 2018*) and local human livelihoods (*Niang et al., 2014*).

However, evidence for changes in forest function linked to weather conditions in equatorial Africa is extremely rare, mainly due to missing long-term meteorological data. The number of rain gauge stations reporting data across Central Africa fell from a peak of more than 50 between 1950 and 1980 to fewer than 10 in 2010 (*Washington et al., 2013*). This low density of observations and poor understanding of local landscape and climatic processes (*Nicholson & Grist, 2003*) limits the accuracy of gridded observational data products (*Asefi-najafabady & Saatchi, 2013*; *Suggitt et al., 2017*). Uncertainty is particularly high for rainfall patterns, which unlike temperature, are poorly conserved over space (*Habib, Krajewski & Ciach, 2001*; *Kidd et al., 2017*). Because of missing ground data, climate and ecological models rely heavily on satellites despite major issues with this data source that include extreme cloudiness in the region and little opportunity for ground-truthing (*Washington et al., 2013*; *Maidment et al., 2014*; *Wilson & Jetz, 2016*; *Dommo et al., 2018*). Empirical meteorological data are urgently needed to verify regional climate and vegetation models and shed light on the mechanisms that drive seasonal and long-term climatic variability in tropical Africa (*Guan et al., 2013*; *Abernethy, Maisels & White, 2016*).

We have the rare opportunity to analyse a 34 year record of rainfall and temperature (and shorter periods of humidity, wind speed, solar radiation and aerosol optical depth) from a long-term ecological research site in western equatorial Africa. These local

weather data have not contributed to the available climate products (such as the high-resolution gridded dataset from the Climate Research Unit) and are able to act as an independent control. In this article we briefly review the published literature on drivers of weather variability and long-term climate trends in western equatorial Africa (~6°S–5°N, 8°–18°E, covering Cameroon, Republic of Congo, Central African Republic, Democratic Republic of Congo, Equatorial Guinea and Gabon). We then use our ground data to analyse seasonal, inter-annual and long-term weather patterns in this data-poor region with particular focus on rainfall for which uncertainty in regional products is high.

## Seasonality

The climate of equatorial Africa is characterised by a bimodal rainfall pattern. Two rainy seasons occur each year coinciding with the boreal spring and autumn when the sun passes directly over the equator (March–May and October–November). Just 3% total annual rainfall falls during the major dry season, which extends from June to August/September (*Balas, Nicholson & Klotter, 2007*). The primary influence on equatorial rainfall has historically been understood to be the Inter Tropical Convergence Zone (ITCZ), a band of clouds and high precipitation that migrates northwards and southwards over the equator following the sun (*Nicholson, 2018*; Fig. 1). However recent developments show the ITCZ to be a poor explanation of seasonal rainfall in Africa, with ITCZ-associated low-level convergence often decoupled from the rain belt in western and central equatorial regions (*Nicholson, 2018*). Improved mechanistic models of the seasonal evolution of atmospheric conditions in the region are urgently needed.

In western equatorial Africa the rainy seasons coincide with bright conditions. Convection clouds develop into storms late in the day or night leaving clear skies during the daytime (*Gond et al., 2013*). By contrast, light is least available during the long dry season due to persistent low-lying cloud cover throughout the day (*Philippon et al., 2019*). The seasonal synchrony between light and precipitation in western equatorial Africa is in contrast to the central Congo Basin and the neotropics where dry seasons tend to coincide with peak irradiance (*Wright & Calderón, 2018*; *Philippon et al., 2019*). In western equatorial Africa the long dry season is also the coolest time of year (*Munzimi et al., 2015*; *Tutin & Fernandez, 1993*).

## Oceanic influences

Large-scale patterns in sea surface temperatures (SSTs) are known to influence weather conditions across the tropics (*Camberlin, Janicot & Poccard, 2001*; Fig. 1). The El Niño Southern Oscillation (ENSO) refers to the state of the atmosphere and surface temperatures of the tropical Pacific Ocean. ENSO has a relatively straightforward, instantaneous effect on temperature throughout the African continent, with greater warming in El Niño years (*Collins, 2011*). Central African rainfall is also strongly connected to SSTs (*Otto et al., 2013*), although interactions are complex and seasonally specific. In Table 1 we summarise six major studies of ocean influences on rainfall in western equatorial Africa (*Todd & Washington, 2004*; *Balas, Nicholson & Klotter, 2007*; *Otto et al., 2013*; *Preethi et al., 2015*; *Nicholson & Dezfuli, 2013*; *Dezfuli & Nicholson, 2013*).

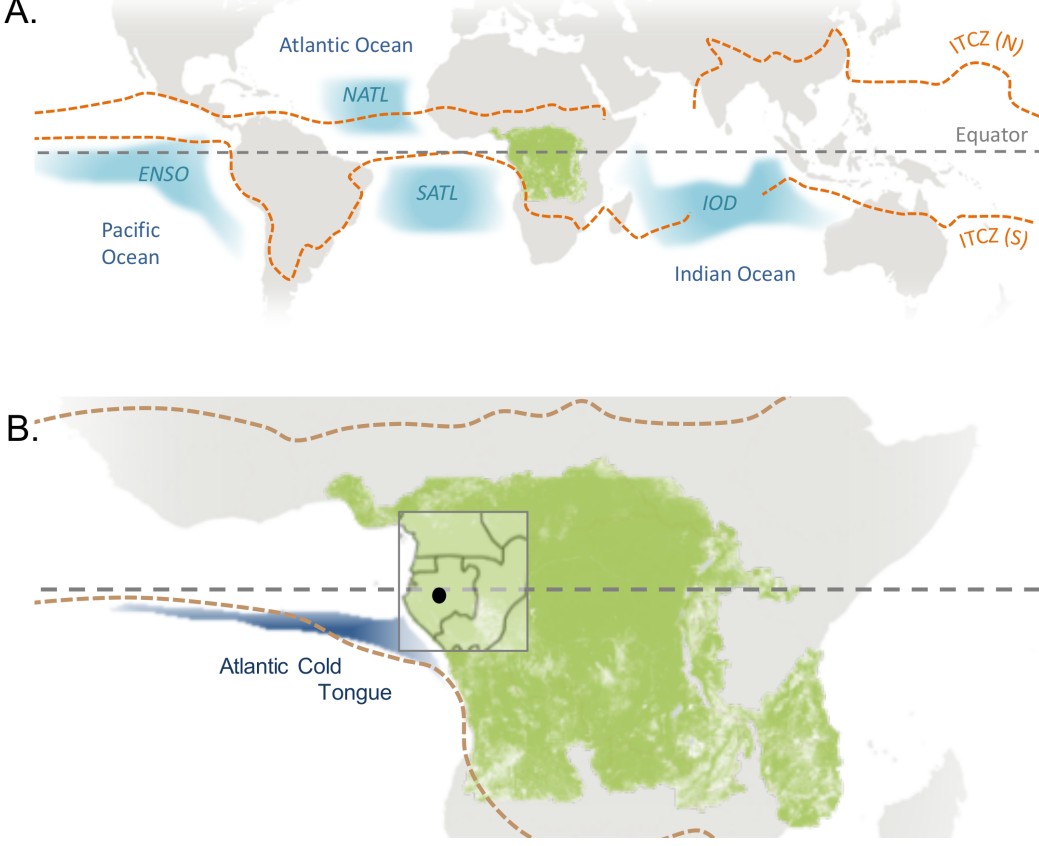

**Figure 1 Global climatic influences on western equatorial Africa.** (A) The forested region of central Africa is indicated by a layer of green pixels (>25% tree cover in 2000 from *Hansen et al. (2013)*—available from http://earthenginepartners.appspot.com/science-2013-global-forest). The Northern (July) and Southern limits (January) of the Inter Tropical Convergence Zone (ITCZ) are drawn from *Barlow et al. (2018)*. The blue zones indicate patterns in oceanic sea surface temperatures (SSTs) known to influence weather in western Central Africa: the Pacific Ocean El Niño Southern Oscillation (ENSO); North and South Tropical Atlantic SSTs (NATL and SATL) and the Indian Ocean Dipole (IOD). In conventional El Niño years the tropical Eastern Pacific is abnormally warm, in El Niño Modoki the warming occurs in the central Pacific. The IOD is the difference between SSTs of the western and eastern tropical Indian Ocean. (B) Lopé National Park (our study site) is indicated by a black dot and the limits of western equatorial Africa as defined in this article are indicated by the grey rectangle (including the humid forests of Gabon, Equatorial Guinea, Cameroon and the Republic of Congo). Also shown is the location of the seasonal Atlantic cold tongue, a pool of cool surface water that develops in the eastern tropical Atlantic during the boreal summer (drawn from *Tokinaga & Xie (2011)*). The grey world map was created by Layerace at Freepik.com.

The main agreements between these studies are that (1) rainfall is below average from February to August in El Niño years (*Camberlin, Janicot & Poccard, 2001*; *Todd & Washington, 2004*; *Balas, Nicholson & Klotter, 2007*; *Preethi et al., 2015*; *Nicholson & Dezfuli, 2013*), (2) rainfall positively correlates with the temperature of the Indian Ocean in January and February (*Balas, Nicholson & Klotter, 2007*; *Preethi et al., 2015*) and (3) warm SSTs in the tropical south Atlantic enhance rainfall from April to September (*Camberlin, Janicot & Poccard, 2001*; *Balas, Nicholson & Klotter, 2007*; *Otto et al., 2013*;

**Table 1 Major oceanic influences on rainfall in western equatorial Africa.**

| Study | Description | Ocean influences | |
|---|---|---|---|
| *Preethi et al. (2015)* | Africa-wide; Satellite and gridded obs.; 1979–2010 | Pacific: | Canonical El Niño reduces rainfall January–September El Niño Modoki increases rainfall March–May |
| | | Indian: | Positive relationship between SSTs and rainfall January–February No relationship between IOD and rainfall |
| *Camberlin, Janicot & Poccard (2001)* | Sub-Sahara; Gridded obs.; 1951–1997 | Pacific: | El Niño negatively influences rainfall April–June |
| | | Atlantic: | South Atlantic SSTs positively influence rainfall April–September |
| *Balas, Nicholson & Klotter (2007)* | WEA; Precipitation gauge dataset; 1950–1998 | Pacific: | El Niño negatively influences rainfall |
| | | Indian: | Weak positive relationship between SSTs and rainfall in all seasons except March–May when it is reversed |
| | | Atlantic: | Positive correlation between south Atlantic SSTs and rainfall Jun-Nov, negative influence December–February Benguela coast influences rain March–May |
| *Todd & Washington (2004)* | CEA and WEA; Gridded obs. and discharge data February–April; 1901–1998 | Pacific: | El Niño has weak negative influence on rainfall February–April |
| | | Atlantic: | North Atlantic Oscillation negatively influences rainfall February–April |
| *Otto et al. (2013)* | CEA and WEA; Simulated data. Dry seasons only | Pacific: | ENSO influences rainfall in dry seasons |
| | | Indian: | IOD negatively influences rainfall in dry seasons |
| | | Atlantic: | Warm tropical Atlantic SSTs enhance rain in dry seasons |
| *Nicholson & Dezfuli (2013)* and *Dezfuli & Nicholson (2013)* | WEA. Regionalised obs. rainy seasons only | Pacific: | El Niño reduces rainfall in rainy seasons |
| | | Indian: | Positive IOD modes associated with reduced rainfall in rainy seasons |
| | | Atlantic: | Warm tropical Atlantic SSTs enhance rainfall in rainy seasons. Strong correlation with Benguela coast from October–December |

Note:
CEA, central equatorial Africa; WEA, western equatorial Africa; SST, sea surface temperatures; ENSO, El Niño Southern Oscillation; IOD, Indian Ocean Dipole.

*Nicholson & Dezfuli, 2013*). We found no evidence in the literature for the influence of large-scale climate oscillations on other weather variables such as light availability, wind speeds or aerosols in the region.

## Long-term trends

There is high confidence in the evidence for warming over African land regions (*Niang et al., 2014*). Satellite estimates for tropical Africa show an annual mean temperature increase of 0.15 °C per decade from 1979 to 2010 (*Collins, 2011*). A recent multi-model ensemble shows that mean temperature for the whole continent is likely to continue to increase more than the global average especially in the long dry season (*James & Washington, 2013*).

Tropical land areas globally have seen no overall change in precipitation over the last century, with a recent increase in precipitation (2003–2013) reversing a drying trend from the 1970s to the 1990s (*Hartmann et al., 2013*). Rainfall patterns are poorly conserved spatially and conflicting trends are detected within the western equatorial region of Africa. A regionalised long-term dataset for Africa constructed from historical records and rain gauge observations shows a sharp reduction in rainfall in the Cameroon region from the late 1960s until the present and a contrasting wetting trend in the Congo/Gabon region from 1980s until the present (*Nicholson, Funk & Fink, 2018*). However a higher resolution analysis of the same dataset shows that within central Gabon there has been a drying trend from the 1970s until 2000 and that there is no data originating from this area for the last two decades (*Nicholson, Funk & Fink, 2018*). Flow data for the river Ogooué—the largest river in western equatorial Africa—indicates that runoff in the region declined from the 1960s until 2010 and that the flood peak has moved from May to April (*Mahe et al., 2013*). Land-cover change has been minimal in the watershed during this period (*Abernethy, Maisels & White, 2016*) and so it is likely that reduced rainfall has been the biggest influence on flow reduction.

Predictions of future rainfall vary widely across the African continent with high uncertainty in the direction of change centrally due to the sparse network of observations and poor understanding of local climate forcing (*James & Washington, 2013*). Model projections mostly show no change or a weak wet signal in the central Congo Basin and a dry signal in the western region in scenarios where warming is greater than 2 °C (*James, Washington & Rowell, 2013*). Models that support a drying trend in western equatorial Africa show strong associations with Atlantic and Indian (but not Pacific) SSTs. The construction of these dry models suggests that reductions in rainfall in Gabon and surrounding countries are likely to be caused by a northward displacement of the equatorial rain belt associated with the Atlantic cold tongue (Fig. 1B) and an eastward shift in convection caused by contrasts between Indian and Atlantic SSTs (*James, Washington & Rowell, 2013*).

As for surface solar radiation, once again the picture varies spatially within central Africa. In the central Congo Basin (14E–30E) there has been a recent widespread decline in cloud optical thickness and no change in aerosol optical thickness (MODIS, 2000–2012) leading to an increase in downward photosynthetically available radiation (CERES, 2003–2012; *Zhou et al., 2014*). For sunshine duration, there has been no change in the central region but a weak decline (2–4 h per decade) in western equatorial Africa from 1983 to 2015 (SARAH-2, *Kothe et al., 2017*). There is no published data on long-term changes in relative humidity or wind speed in the region.

## Implications for the eco-region

Humid evergreen forests currently dominate western equatorial Africa despite relatively low precipitation compared to other closed canopy tropical forests (*Reich, 1995*). As summarised by *James, Washington & Rowell (2013)*, intense rainfall seasonality alongside a drying and warming climate is likely to lead to water stress and could push these ecosystems towards more open, fire prone, dry forest systems (as evidenced in this

region over the last 3,000 years; *Brncic et al., 2006* and in West Africa in recent decades: *Fauset et al., 2012*) or even savanna (*Willis et al., 2013*). While elevated atmospheric carbon dioxide improves water use efficiency, the balance of gains for tropical forests related to carbon fertilisation vs. losses related to further warming and drying are poorly characterised in climate and vegetation models (*Huntingford et al., 2013*). Tropical forest species turnover or loss related to climate change will have serious consequences for people and animals dependent on forest resources in the region (*Abernethy, Maisels & White, 2016*) while loss of tree cover will impact carbon storage and even feedback onto climate (*Mitchard, 2018*). Despite the risks associated with these scenarios few meteorological data are available, especially in recent decades, to understand if the climatic trends described above are witnessed on the ground and how quickly they are progressing. Using ground data from Lopé National Park (NP), Gabon, collected over a 34-year period we ask: how fast is the region warming? Is the region drying and how quickly? And how do the oceans influence rainfall and temperature variability? Answers to these questions will be important to predict the viability of evergreen forest ecosystems under future climates.

## MATERIALS AND METHODS

### Description of the study area and weather data recorded since 1984

The Station d'Études des Gorilles et Chimpanzées (SEGC) research station is located at the northern end of Lopé National Park, Gabon (−0.2N, 11.6E). The station sits in a tropical forest-savanna matrix, at an elevation of 280 m and within 10.5 km of the river Ogooué (the largest river in Gabon and the country's main watershed). Ecological research activities including weather, plant and animal observations have taken place continuously at SEGC from 1984 until the present (>300 publications; 1984–2018).

Weather data have been recorded at Lopé using various types of equipment at two locations: a savanna site (the research station; 11.605E, −0.201N) and a forest site (800 m from the research station and approximately 10 m from the savanna/forest edge; 11.605E, −0.206N; Table 2). From 1984 to the present, a manual rain gauge was placed at the savanna site (50 cm above ground and >5 m from any tree or building) and used to record total daily rainfall at 8 AM each morning. There was a gap in data recording in 2013 and occasional missing days due to logistical constraints (e.g. availability of personnel). Since 1984 daily maximum and minimum temperatures and relative humidity were recorded using a manual thermometer and wet/dry bulb located at the forest site (1.5 m aboveground under closed canopy), which were checked whenever field teams passed it or daily when logistics permitted. In 2002 all temperature recording at the forest site was transferred to continuous automatic units (ONSET HOBO® Data Loggers ref https://www.onsetcomp.com/, these units also recorded relative humidity). At the same time temperature recording using the HOBO units also began in the savanna. Due to technical failures these units were replaced in 2006 with the original manual max/min thermometer in the forest and a digital max/min thermometer (Taylor, 1441) in the savanna. These were in turn replaced by another type of automated unit (TinyTag Plus 2, Gemini Data Loggers https://www.geminidataloggers.com/data-loggers/tinytag-plus-2, some of which record both temperature and relative humidity). TinyTags were deployed in

**Table 2 Weather station instrument record at Lopé NP, Gabon, 1984–2018.**

| Instrument | Time period | Location | Data | Missing periods |
|---|---|---|---|---|
| Manual rain gauge | 1984–present | Savanna | Total daily rainfall | September-2010 to December 2010; 2013; Odd days |
| Manual max/min thermometer | 1984–2002; 2006–2007 | Forest | Max./min. temp. since last reset | July-1998 to January-1999; March-2001 to August-2001; Intermittent throughout |
| Wet/dry bulb | 1984–2002 | Forest | Relative humidity | Intermittent throughout |
| HOBO Data Logger (ONSET) | 2002–2006 | Forest + Savanna | Temperature relative humidity | June-2003 |
| Digital max/min thermometer (Taylor 1441) | 2006–2008 | Savanna | Max./min. daily temp | Odd days |
| TinyTag Plus 2 Data Logger (Gemini) | 2007–present | Forest + Savanna | Temperature; relative humidity | Jun-2015 to Jun-2016 (Forest); Intermittent through 2017 |
| Vantage Pro2 Weather Station (Davis) | 2012–2014 | Savanna | Rainfall; temperature; relative humidity; pressure; wind speed; wind direction; UV index; solar radiation | November-2013; February-2014 to July-2014 |
| Minimet Weather Station (SKYE) | 2013–2016 | Savanna | Temperature; relative humidity; wind speed; wind direction; solar radiation | January-2014 to November-2014; Intermittent throughout |
| Sun Photometer (NASA Aeronet) | 2014–present | Savanna | Aerosol optical depth | Intermittent throughout |

the forest from 2007 and in the savanna from 2008 and used until the present (with a gap at the forest site from mid-2015 to mid-2016 and intermittent recording throughout 2017 partly due to equipment malfunctions caused by termite infestation). Two weather stations were installed in the savanna (sited near the research station, on a rock 4 m from the ground) and collected data between 2012 and 2016. A Davis VantagePro2 (https://www.davisinstruments.com/solution/vantage-pro2/) was installed in January 2012 and recorded rainfall, temperature, relative humidity, pressure, wind speed and direction, UV index and solar radiation every 30 min for 2 years until the equipment was struck by lightning in January 2014. A SKYE MINIMET weather station (https://www.skyeinstruments.com/minimet-automatic-weather-station/) was installed at the same location in 2013 and collected temperature, relative humidity, wind speed and direction and solar radiation (but not rainfall as the gauge was defective). The SKYE unit ran intermittently until 2016 when the equipment was also damaged by lightning: data records between January 2014 and November 2014 were also lost. Finally, a sun photometer was installed at the research station in April 2014 and used to record aerosol optical depth up to the present as part of the NASA Aerosol Robotic Network (Aeronet; https://aeronet.gsfc.nasa.gov/; *Holben et al., 1998*).

Despite sustained effort, the remote and challenging environment at Lopé has led to a patchy weather data record. This situation has been exacerbated since the introduction of automated loggers, due to unreliable performance combined with difficulties and time delays in replacing or repairing malfunctioning equipment and respecting annual

calibration schedules with manufacturers based in Europe or the USA. New equipment was often introduced out of necessity when previous equipment failed, precluding the opportunity of collecting simultaneous data for standardisation. Such problems have been experienced at many other field stations across Africa (*Maidment et al., 2017*) and homogenisation is necessary in most long-term instrumental climatic data sets (*Peterson et al., 1998*). It was therefore necessary to select and standardise the Lopé data to reduce systematic biases between recording equipment. We summarise the data selection steps we undertook below and provide further detail in the accompanying Supplemental Information (Article S1 and Code S1). All Lopé data can be downloaded from the University of Stirling's DataSTORRE (http://hdl.handle.net/11667/133).

## Data cleaning and preparation

We constructed a long-term record of daily rainfall totals (1984–2018) by calibrating the two sources of data (manual rain gauge and Vantage Pro weather station) using a simple linear model on simultaneous records and taking the mean value for days with multiple observations (resulting in a dataset of 12,050 complete daily observations out of a possible 12,419 over 34 years). Where possible we interpolated missing daily values using the 10-day running mean for the time series (resulting in a dataset of 12111 interpolated daily observations), however 11 months spread over three calendar years remained incomplete. We used these interpolated daily data to calculate total monthly and annual rainfall for the months and years with complete data (397 complete monthly observations out of a possible 408 and 31 complete years out of a possible 34).

Temperature data were recorded using six different types of equipment across two sites (recorded in the forest from 1984 to 2018 and in the savanna from 2002 to 2018). Where there were multiple observations from overlapping data records we calculated mean daily maximum and minimum values for each site and day in the time series and used this dataset to demonstrate temperature seasonality at each site (resulting in a dataset of 7,058 daily observations out of a possible 12,419 over 34 years at the forest and 4,878 daily observations out of a possible 5,844 over 16 years at the savanna). To create continuous time series for periodicity analyses we calculated mean monthly maximum and minimum daily temperatures for each month in the time series with more than five observations (resulting in a dataset of 327 monthly observations out of a possible 408 from the forest site and 166 monthly observations out of a possible 192 at the savanna site). Minimum daily temperatures are recorded during the night and thus avoid errors associated with direct solar radiation (which we found to vary between our equipment, Article S1). Because of this we chose to use minimum daily temperatures to assess long-term trends and inter-annual variation. We constructed a long-term daily record by calculating mean daily minimum temperature using data from both sites combined (8,217 daily observations out of a possible 12,419 over 34 years). We summarised these data to a monthly mean time series for months with more than five observations (372 monthly observations out of a possible 408 over 34 years).

Finally, we used the shorter (and/or patchier) periods of data available for relative humidity (2002–2018), solar radiation (2012–2016), wind speed (2012–2016) and aerosol

optical depth (2014–2017) to assess seasonality and periodicity for these climate variables. We used night-time relative humidity records (6 PM–6 AM) to avoid errors associated with direct solar radiation and converted to absolute humidity (g/m$^3$) using simultaneous temperature records within the R package *humidity* (*Cai, 2018*). We extracted aerosol optical depth data at wavelengths relevant for photosynthetic activity (440, 500 and 675 nm).

## Gridded regional temperature datasets

Because of missing data and lack of simultaneous recording between temperature equipment at Lopé we also downloaded two widely used gridded regional data products with which to compare the Lopé data: daily minimum air temperature from the Gridded Berkeley Earth Surface Temperature Anomaly Field (1° resolution; *Rohde et al., 2013*) and monthly mean daily minimum temperature from the Climate Research Unit's Time-Series v4.01 of high-resolution gridded data (CRU TS4.01; 0.5° resolution; *University of East Anglia Climatic Research Unit, Harris & Jones, 2017*; *Harris et al., 2014*). Both were downloaded from http://climexp.knmi.nl/start.cgi for the grid-cell overlapping the SEGC location (0.2N, 11.6E).

## Ocean sea surface temperatures

We downloaded data for four oceanic SST indices from commonly used data sources: the Multivariate ENSO Index (MEI; *Wolter & Timlin, 1993*, *1998*) sourced from the NOAA website (https://www.esrl.noaa.gov/psd/enso/mei/index.html), the Indian Ocean Dipole (IOD) Dipole Mode Index (*Saji & Yamagata, 2003*) sourced from the NOAA website (https://www.esrl.noaa.gov/psd/gcos_wgsp/Timeseries/DMI/) and deseasonalised SSTs for the tropical north Atlantic (NATL, 5°–20°N, 60°–30°W) and the south equatorial Atlantic (SATL, 0°–20°S, 30°W–10°E) sourced from the NOAA National Weather Service Climate Prediction Center (http://www.cpc.ncep.noaa.gov/data/indices/). We rescaled all four SST indices by subtracting the mean and dividing by one standard deviation to allow direct comparison of their effects. Positive values for MEI indicate El Niño conditions; positive values for NATL and SATL indicate warm SSTs in those regions while positive values for IOD indicate cool SSTs in South Eastern equatorial Indian Ocean and warm SSTs in the Western equatorial Indian Ocean.

## Analyses

### Seasonality

To characterise the seasonality of each weather variable we calculated mean values from empirical daily data at three different scales: the mean value for each day of the calendar year (DOY, fine-scale), the 10-day running mean of DOY (medium-scale) and the mean value for each calendar month (coarse-scale). To formally assess the periodicity of each variable we used Fourier analysis. The Fourier transform is a form of spectral analysis used to calculate the relative strength of all possible regular cycles in time series data (*Bush et al., 2017*). We created standardised, complete time series by filling missing values in monthly time series using the mean value for the corresponding calendar

month and standardising the data by subtracting the mean and dividing by its standard deviation. We then computed the Fourier transform for each time series using the *spectrum* function from the R Stats package (*R Core Team, 2019*) and inspected the spectra plots for peaks that represent strong regular cycles in the data (*Bush et al., 2017*).

### Long-term trends

We used a linear regression framework to test whether rainfall and minimum temperature had changed over the observation period (1984–2018) using non-interpolated daily data. We fitted compound Poisson generalised linear mixed models (CPGLMM) for daily rainfall and linear mixed models (LMM) for minimum daily temperature to account for their respective data distributions. CPGLMMs are exponential dispersion models based on the Tweedie distribution and are recommended for daily or monthly rainfall data which is positive and continuous with many exact zeros (*Hasan & Dunn, 2010*). We fitted CPGLMMs using the *cplm* R package (*Zhang, 2013*) and LMMs using the *lme4* R package (*Bates et al., 2015*). DOY was included as a random intercept in all models to account for seasonality and the hierarchical structure of the data. We fitted initial models with Year (continuous, rescaled) as the predictor (representing long-term change) and compared these to intercept-only models (representing no long-term change) preferring simple models (few parameters) with lowest AIC (significantly different if delta AIC >2). See R-style model notation below with $\varepsilon$ representing residual error not accounted for by the predictors of the model.

1. Daily Rainfall ~ Year + (1|DOY) + $\varepsilon$
2. Daily Rainfall ~ 1 + (1|DOY) + $\varepsilon$
3. Minimum Daily Temperature ~ Year + (1|DOY) + $\varepsilon$
4. Minimum Daily Temperature ~ 1 + (1|DOY) + $\varepsilon$

We repeated the same procedure for gridded temperature data for Lopé from the daily Berkeley and monthly CRU datasets. DOY was included as a random intercept within the models with daily response data and Month was included as a random intercept within the models with monthly response data.

5. Minimum Daily Temperature (Berkeley) ~ Year + (1|DOY) + $\varepsilon$
6. Minimum Daily Temperature (Berkeley) ~ 1 + (1|DOY) + $\varepsilon$
7. Mean Monthly Minimum Daily Temperature (CRU) ~ Year + (1|Month) + $\varepsilon$
8. Mean Monthly Minimum Daily Temperature (CRU) ~ 1 + (1|Month) + $\varepsilon$

Next we investigated whether trends in rainfall and minimum temperature at Lopé differed by season. Various seasonal definitions are used throughout the tropics, usually related to the annual rainfall cycle. We defined our seasons according to Lopé rainfall climatology where the long dry season extends into September, that is October–November (ON, the short rainy season), December–February (DJF, the short dry season), March–May (MAM, the long rainy season) and June–September (JJAS, the long dry season; Fig. 2A). We included Year (continuous, rescaled), Season (factor with four levels

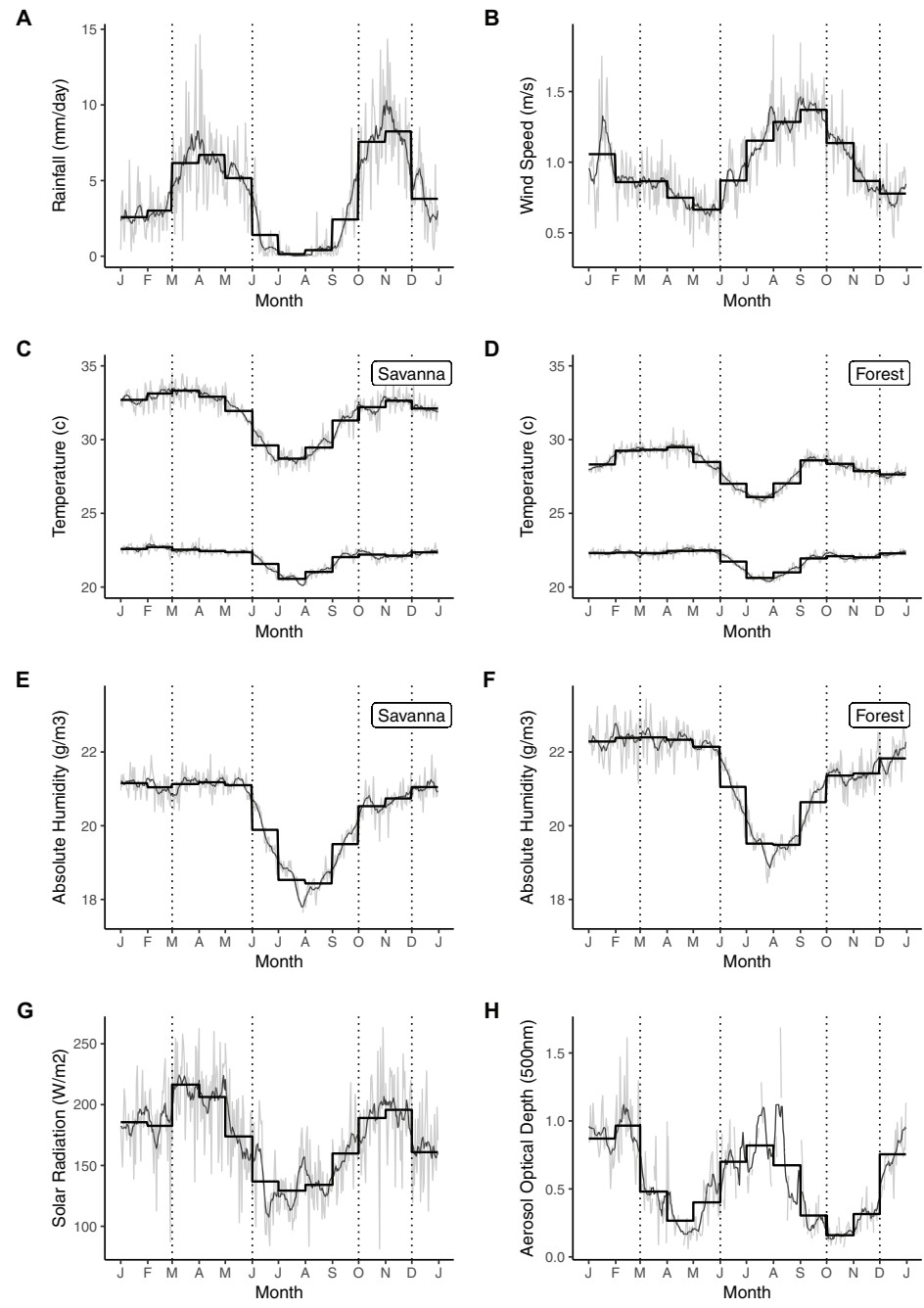

**Figure 2 Seasonal weather variability at Lopé NP, Gabon.** Mean seasonality for (1) daily rainfall (1984–2018), (B) wind speed (2012–2016), (C and D) minimum and maximum temperatures in the savanna (2002–2018) and forest (1984–2018), (E and F) absolute humidity in the savanna and forest (2007–2015), (G) surface solar radiation (2012–2016) and (H) aerosol optical depth at 500 nm (2014–2017). The thin grey lines indicate the mean values for each day of the calendar year (DOY). The thin black lines indicate the 7-day running means of DOY and the thick black lines indicate the monthly means. Vertical dotted lines indicate the alternating rainy and dry seasons.

as above) and their interaction as predictors in initial models to represent long-term change varying by season. We fitted subsequent models without the interaction term to represent long-term change not varying by season and compared the models using AIC values. DOY was included as a random intercept in all models, as before.

9. Daily Rainfall ~ Season + Year + Season: Year + (1|DOY) + ε
10. Daily Rainfall ~ Season + Year) + (1|DOY) + ε
11. Minimum Daily Temperature ~ Season + Year + Season: Year + (1|DOY) + ε
12. Minimum Daily Temperature ~ Season + Year + (1|DOY) + ε

To estimate the magnitude of the trend in each season, rather than comparing to the global intercept, we modified the best models by temporarily removing the global intercept. For all models described above we inspected the residuals to check for temporal autocorrelation using the R package *itsadug* (*Van Rij et al., 2017*). None of the median autocorrelation functions (autocorrelation calculated for each DOY or Month respectively) showed significant temporal autocorrelation.

### Periodicity over time

We used Wavelet analyses to assess if and how the periodicities of the rainfall and temperature time series have changed over time. The Wavelet transform extends the Fourier transform into the time-frequency domain and allows identification of cyclic behaviour that may be transient or change over time (*Torrence & Compo, 1998*). We used the complete, standardised monthly time series for rainfall and minimum temperature (with missing values interpolated from the long-term calendar month mean) and computed the Wavelet transform using the function *wt* from the R package *biwavelet* (*Gouhier, Grinsted & Simko, 2018*). From the wavelet transform we plotted the power (higher power denotes greater fidelity to a certain cycle), significance (a cycle is significant if >0.95, $X^2$ test) and cone of influence (denoting the unreliable region at the beginning and end of the time series due to edge effects). We extracted the power of the biannual, annual and multiannual (mean of the 2–4 year periods) components from the wavelet spectra to further assess how these dominant cycles have varied over time and contributed to the trend (*Adamowski, Prokoph & Adamowski, 2009*). We constrained the upper limit of the multiannual component to 4 years because longer cycles were heavily influenced by edge effects.

### Oceanic influences

We used wavelet coherence to assess if and how the local weather system at Lopé is associated with SSTs of the major oceans at interannual scales (Pacific: MEI, Indian Ocean: IOD and Atlantic Ocean: NATL and SATL). Wavelet coherence is an approach derived from bivariate wavelet analysis and calculates a measure of the correlation (from 0 to 1) between two time series (*x* and *y*) at all periodicities through time. Wavelet coherence can be used to identify common oscillatory behaviour, even if that behaviour is inconsistent (i.e. the time series are 'non-stationary;' *Grinsted, Moore & Jevrejeva, 2004*). According to *Grinsted, Moore & Jevrejeva (2004)*, strong coherence and consistent phase

relationships between two carefully selected time series indicate that there may be a causative relationship. In this study we computed wavelet coherence for all eight combinations of *x* (rainfall or minimum temperature monthly time series) against *y* (MEI, IOD, NATL or SATL monthly time series) using the function *wtc* from the R package *biwavelet* (*Tarik, Aslak & Viliam, 2019*) with 1000 Monte Carlo randomisations. To summarise and compare the wavelet coherence between each time series pair we calculated the 'global' time-averaged coherence for each period (*Chang et al., 2019*).

R code to accompany all analyses described above is made available in Supplemental Information (Code S1). Permission to conduct this research in Gabon was granted by the Centre International de Recherches Medicales de Franceville (CIRMF) Scientific Council and the Ministry of Water and Forests (1986–2010), and by Gabonese National Parks Agency (ANPN) and the National Centre for Research in Science and Technology (CENAREST; 2010-present).

# RESULTS

## Seasonality

Mean total annual rainfall at Lopé from 1984 to 2018 was 1,466 mm ± 201 SD. Rainfall in this period followed a biannual cycle (Fig. S1) with broad peaks in the rainy seasons (MAM and ON) when mean daily rainfall was always greater than 5 mm (Fig. 2A). The long dry season (JJAS) was very consistent, with a 90-day period (mid-June to mid-September) in which the 10-day running mean was never greater than 1 mm (Fig. 2A). The short dry season (DJF) by contrast was much less dry (10-day running mean greater than 1 mm) and more variable between years (Fig. 2A).

Mean daily maximum and minimum temperatures at Lopé were 28.1 °C ± 2.2 SD and 21.9 °C ± 1.1 SD respectively at the forest site (1984–2018) and 31.6 °C ± 2.9 SD and 22.0 °C ± 1.2 SD at the savanna site (2002–2018). Daily temperature range was greater in the savanna than under the forest canopy (Figs. 2C and 2D). Maximum daily temperature in the forest showed strong annual and bi-annual cycles while in the savanna only the annual cycle appeared dominant (Fig. S1). The difference between the two sites occurred during the short dry season when temperatures were maintained in the savanna at similar levels to the rainy seasons (10-day running mean always greater than 31.7 °C from October to May in the savanna; Fig. 2C). In the forest, the highest peaks in maximum daily temperature occurred in April and September (mean monthly maximum daily temperatures were 29.5 °C and 28.6 °C respectively; Fig. 2D). Annual cycles dominated the minimum daily temperature record for both the forest and the savanna (Fig. S1). Minimum daily temperatures were relatively constant from September to June (~22.5 °C) followed by a cool period during the long dry season reaching an annual trough in July (mean monthly minimum daily temperature is 20.6 °C in both the savanna and forest; Figs. 2C and 2D).

The forest was more humid than the savanna throughout the year (mean absolute humidity is 21.40 g/m$^3$ and 20.35 g/m$^3$ respectively; Figs. 2E and 2F). Humidity follows the same annual cycle in both locations (Fig. S1), dropping during the long dry season to reach

**Table 3 Model comparisons to test for long-term trends in rainfall and minimum temperature at Lopé NP, Gabon (1984–2018).** We used a compound poisson generalised linear mixed model for daily rainfall and a linear mixed model for minimum daily temperature. Day of Year was included as a random intercept in both models.

| Response | Model | Predictors | DF | AIC | Delta AIC |
|---|---|---|---|---|---|
| Rainfall | Long-term change | Year | 4 | 40,839.6 | 0.0 |
| | No long-term change | Intercept only | 3 | 40,842.3 | 2.7 |
| Temperature | Long-term change | Year | 4 | 22,909.5 | 0.0 |
| | No long-term change | Intercept only | 3 | 23,494.0 | 584.5 |

**Note:**
    AIC, Akaike Information Criterion; DF, Degrees of Freedom.

a minima in August and increasing throughout the short rains (ON) to reach a plateau from January to May (Figs. 2E and 2F).

Both surface solar radiation and wind speed were dominated by annual cycles at Lopé (Fig. S1), with the long dry season coinciding with low irradiance (mean monthly solar radiation for July = 129.3 W/m$^2$; Fig. 2G) and elevated wind speeds (mean monthly wind speeds for August and September are 1.3 m/s and 1.4 m/s respectively; Fig. 2B). Aerosol optical depth cycled twice yearly (Fig. S1), being elevated during the dry seasons and suppressed during the rainy seasons (Fig. 2H). In contrast to the solar radiation cycle, which reached its minima during the long dry season (JJAS), the strongest peak in aerosol optical depth occurred in the short dry season (mean monthly aerosol optical depth at 500 nm for February = 0.97). Aerosol optical depth at 440 and 675 nm wavelengths is similar to that at 500 nm (Fig. S2).

### Long-term trends

Total annual rainfall decreased by −75 mm per decade, a change of −5.5% relative to mean annual rainfall for the time period (CPGLMM, Estimated index parameter = 1.6, Estimated dispersion parameter = 9.7, Estimate = -0.05, SE = 0.02, Z = −2.22, 96% Confidence Interval = −0.10: −0.01; Table 3 and Fig. 3A). However, the slope of the decline was seasonally dependent (Tables 4 and 5) with no change in daily rainfall in DJF and ON and significant decline in JJAS (−0.07 mm per day per decade, equating to −6.35% of mean JJAS daily rainfall).

Minimum daily temperature at Lopé increased at a rate of +0.25 °C per decade, equivalent to +1.1% relative to mean minimum temperature for the time period (LMM, Estimate = 0.24; SE = 0.01; $T$ = 24.84; 95% Confidence Interval = 0.22: 0.26; Table 3; Fig. 3B). The rate of warming also varied by season (Tables 4 and 5) with minimum temperature increasing most quickly in ON and DJF (+0.31 °C and +0.30 °C per decade respectively) and most slowly in JJAS (+0.18 °C per decade).

Berkeley minimum daily temperature for the interpolated Lopé grid square (1° resolution) increased at a rate of +0.16 °C per decade (LMM, Estimate = 0.34, SE = 0.01, $T$ = 23.4, 95% Confidence Interval = 0.31: 0.37) while the CRU interpolated record (0.5° resolution) increased by +0.19 °C per decade (LMM, Estimate = 0.63 SE = 0.06, $T$ = 11.2, 95% Confidence Interval = 0.52: 0.74).

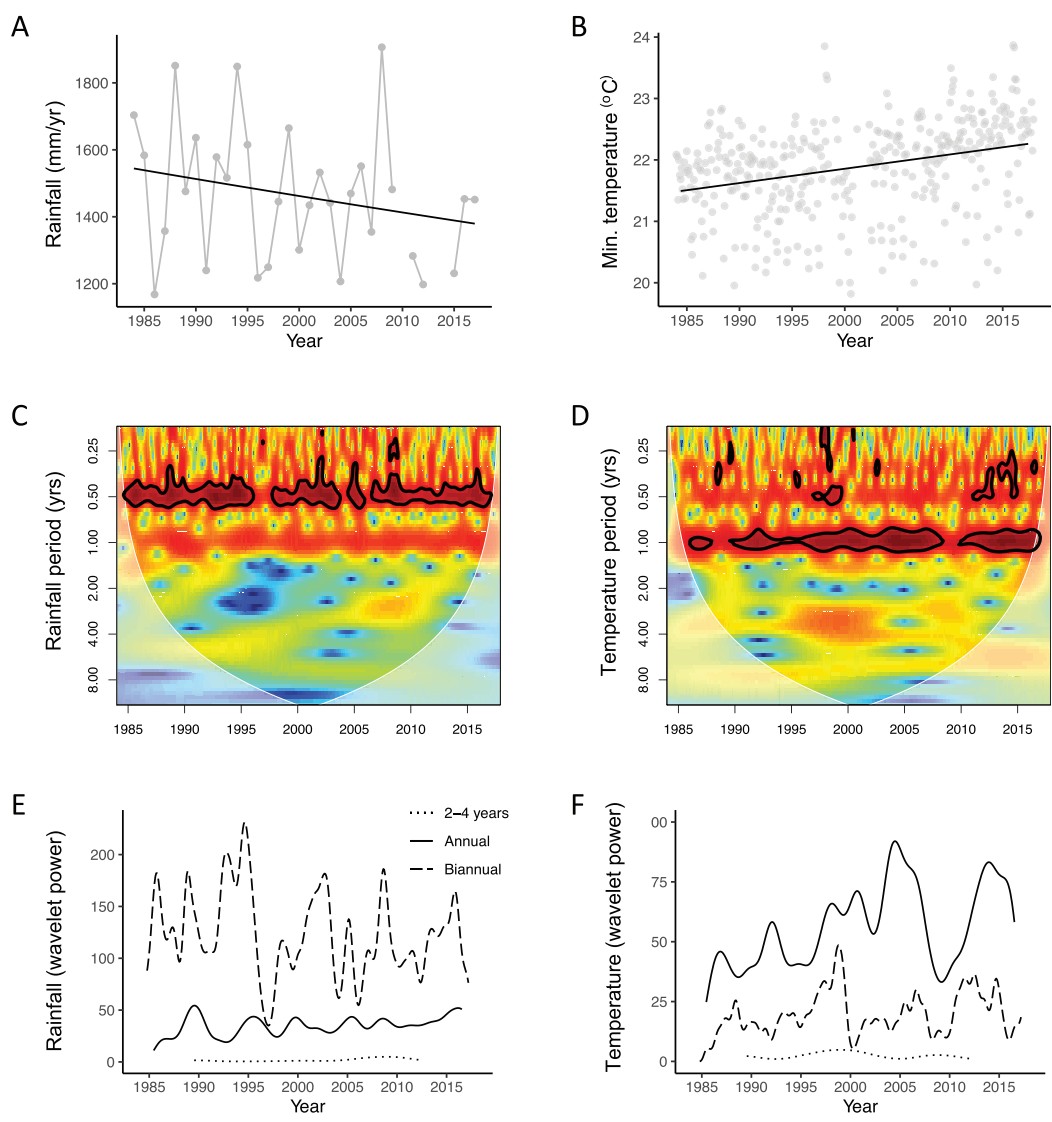

**Figure 3 Inter-annual variation, long-term trends and periodicity for rainfall and temperature at Lopé NP, Gabon.** (A) The grey lines indicate inter-annual variation and the black line indicates the long-term trend for total annual rainfall (1984–2018) derived from a compound poisson generalised linear mixed model. (B) The grey dots indicate raw daily data summarised to monthly means and the black line indicates the long-term trend for minimum daily temperature (1984–2018) derived from a linear mixed model. (C and D) Wavelet transforms of the standardised monthly time-series for total monthly rainfall and mean minimum daily temperature. The faded region indicates the "cone of influence" where end effects make the data unreliable. The colour indicates the power of the cycle at each time period, red, high power and blue, low power. Bold black lines indicate cycles with significant power (Chi-sq test). (E and F) Extracted wavelet components for the biannual, annual and multi-annual (mean of 2–4 years) periods from the wavelet transforms, adjusted for edge effects. Both (E) and (F) share the same legend.

### Periodicity over time

Wavelet analyses gave further indication of the nature of these changes. The dominant six-month cycle for rainfall was, on average, four times as powerful as the annual component and 66 times as powerful as the multi-annual component and remained

**Table 4 Model comparisons to test for long-term trends in rainfall and minimum temperature varying by season at Lopé NP, Gabon (1984–2018).** We used a compound poisson generalised linear mixed model for daily rainfall and a linear mixed model for minimum daily temperature. Day of Year and was included as a random intercept in both models.

| Response | Model | Predictors | DF | AIC | Delta AIC |
|---|---|---|---|---|---|
| Rainfall | Long-term change by season | Year × Season | 10 | 40,506.2 | 0.0 |
| | Long-term change not by season | Year + Season | 7 | 40,519.5 | 13.3 |
| Temperature | Long-term change by season | Year × Season | 10 | 22,572.8 | 0.0 |
| | Long-term change not by season | Year + Season | 7 | 22,582.4 | 9.6 |

Note:
AIC, Akaike Information Criterion; DF, Degrees of Freedom.

**Table 5 Outputs from the best models for long-term trends in rainfall and minimum daily temperature varying by season at Lopé NP, Gabon (1984–2018).** The estimates derive from a compound poisson generalised linear mixed model for daily rainfall and a linear mixed model for minimum daily temperature. Day of Year was included as a random intercept in both models.

| Response | Predictor | Estimate | SE | T | Lower 95% CI | Upper 95% CI |
|---|---|---|---|---|---|---|
| Rainfall | DJF | 0.99 | 0.09 | 10.45 | 0.81 | 1.17 |
| | JJAS | −0.82 | 0.09 | −8.97 | −1.00 | −0.64 |
| | MAM | 1.67 | 0.09 | 18.32 | 1.49 | 1.85 |
| | ON | 1.93 | 0.11 | 17.42 | 1.71 | 2.15 |
| | Year: DJF | 0.03 | 0.05 | 0.62 | −0.07 | 0.13 |
| | Year: JJAS | −0.28 | 0.06 | −5.08 | −0.40 | −0.16 |
| | Year: MAM | −0.06 | 0.04 | −1.38 | −0.14 | 0.02 |
| | Year: ON | 0.00 | 0.05 | −0.06 | −0.10 | 0.10 |
| Temperature | DJF | 22.30 | 0.04 | 534.23 | 22.22 | 22.38 |
| | JJAS | 21.22 | 0.04 | 595.76 | 21.14 | 21.30 |
| | MAM | 22.33 | 0.04 | 542.42 | 22.25 | 22.41 |
| | ON | 21.97 | 0.05 | 433.78 | 21.87 | 22.07 |
| | Year: DJF | 0.30 | 0.02 | 15.13 | 0.26 | 0.34 |
| | Year: JJAS | 0.17 | 0.02 | 10.42 | 0.13 | 0.21 |
| | Year: MAM | 0.25 | 0.02 | 12.92 | 0.21 | 0.29 |
| | Year: ON | 0.30 | 0.02 | 12.33 | 0.26 | 0.34 |

Note:
SE, Standard Error; T, T value; CI, Confidence Interval.

significant for most of the time period (Fig. 3C). However, the signal of the biannual cycle weakened on three occasions (1996–97, 2004 and 2006; Fig. 3C). Over time, the signal of the biannual rainfall cycle appeared to decrease while the annual cycle strengthened (Fig. 3E). The annual cycle for minimum temperature was, on average, three times as powerful as the biannual component and 23 times as powerful as the multi-annual component (Fig. 3F). The signal of the annual cycle remained dominant throughout most of the time period with patches of low power at the end of the 1980s and between 2007 and 2010 (Fig. 3D). There were patches of high power in the multiannual component around 2000. The signal of both the annual and semi-annual components appear to have been increasing in strength over time (Fig. 3F).

*Oceanic influences*

Wavelet coherence analyses showed that the ENSO index (MEI) had the strongest coherence with both rainfall and temperature at Lopé over the last three decades at multi-annual scales (2–4 years; Figs. 4; Fig. S3). However, the influence of ENSO has been patchy through time; Coherence between ENSO and rainfall was particularly strong pre-1990 and between 2007 and 2012 (Fig. 4A) while coherence between ENSO and minimum temperature was fairly consistent pre-2000 and has become weaker since (Fig. 4B). SSTs of the southern tropical Atlantic showed strong coherence with Lopé rainfall pre-2000 while SSTs of the northern tropical Atlantic showed strong coherence with Lopé rainfall post-2000 at multi-annual scales (4–8 years; Figs. 4C and 4E; Fig. S3). SATL cycled in phase with Lopé rainfall (arrows point to the right) while NATL cycled in anti-phase during the 2005–2010 period (arrows point to the left; Figs. 4C and 4E). Within the reliable region of the wavelet coherence plots (away from edge effects) the IOD does not appear to have had a particularly strong or consistent relationship with either rainfall or temperature at Lopé (Figs. 4G and 4H; Fig. S3).

# DISCUSSION

## Our results

Lopé weather has changed significantly over the last three decades, warming at a rate of +0.25 °C per decade (minimum daily temperature) and drying at a rate of −75 mm per decade (total annual rainfall; Figs. 3A and 3B). Both trends are seasonally dependent (Table 4); with significant warming occurring in all seasons, being most pronounced from October to February (see model estimates in Table 5). The rainfall decline occurred predominately between March and September, incorporating both the long rainy season and the long dry season (see model estimates in Table 5). The drying trend at Lopé supports observations of reduced Ogooué river flow from March to September (*Mahe et al., 2013*) and precipitation declines evident from gridded gauge-data for the Gabon/Cameroon region (−1% total annual rainfall, 1968–1998; *Malhi & Wright, 2004*). However, the Lopé total annual rainfall decline of −5.5% per decade exceeds the trend estimated from the regional gauge-data. While the strength of the biannual cycle in rainfall appears to be declining at Lopé along with the overall long-term trend, the annual component is getting more powerful. Declines in rainfall in the long dry season (June–September) but not the short dry season (December–February) are likely to be contributing to an increased contrast between the two dry seasons and enhancing the overall annual rainfall cycle (Table 5).

The warming trend recorded at Lopé is greater than that estimated for the location over the same time period using the Berkeley and CRU gridded datasets (+0.16 °C and +0.19 °C respectively) and that identified using satellite data for mean annual temperature for all tropical Africa (+0.15 °C, 1979–2010; *Collins, 2011*). However, it is lower than the change estimated from gridded observational data (CRU) for mean annual temperature specifically for African tropical forests (+0.29 °C per decade, 1976–1998; *Malhi & Wright, 2004*). While there remain issues with the Lopé temperature data record

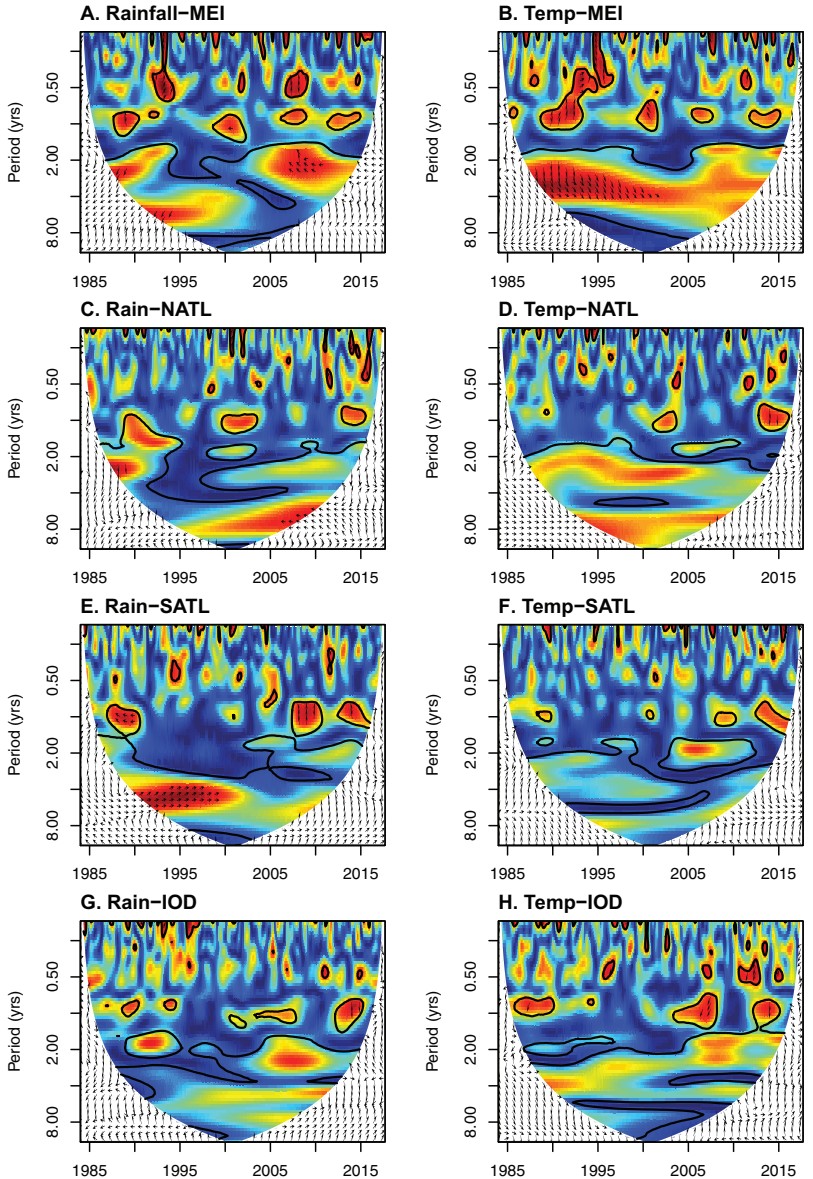

**Figure 4 The influence of oceanic sea surface temperatures on weather at Lopé NP, Gabon.** Wavelet coherence plots for all eight combination of *x* against *y*: (A) Rainfall against the Multivariate ENSO Index (MEI), (B) minimum temperature against MEI, (C) rainfall against northern tropical Atlantic sea surface temperatures (NATL), (D) minimum temperature against NATL, (E) rainfall against southern equatorial Atlantic sea surface temperatures (SATL), (F) minimum temperature against SATL, (G) rainfall against the Indian Ocean Dipole (IOD) and (H) minimum temperature against IOD. The coloured region indicates the reliable data within the "cone of influence" away from edge effects. The colour indicates the strength of coherency between the time series at each period through time, red, high coherency and blue, low coherency. Bold black lines indicate areas with significant coherency (derived from Monte Carlo randomizations). Arrows indicate the phase relationship between the time series within areas of strong coherency. Arrows pointing to the right mean that *x* and *y* are in phase. Arrows pointing to the left mean that *x* and *y* are in anti-phase. Arrows pointing up mean that *y* leads *x* by π/2. Arrows pointing down mean that *x* leads *y* by π/2.

(lack of simultaneous recording to calibrate data recorded using different equipment), there is good evidence from supporting datasets and the literature that the warming trend observed at the site since 1984 is real. The slower warming trend in the already cool, long dry season is likely to account for the apparent increase in the power of the annual cycle for Lopé minimum temperature.

Our analysis of seasonality at Lopé further serves to emphasise the ecological importance of the long dry season in western equatorial Africa; 3–4 months of dry (almost no rainfall for 90 consecutive days), cool (mean maximum daily temperature is 2.5 °C lower in July compared to April) and windy conditions with low humidity and limited light availability (Fig. 2). Such a defined dry season poses specific constraints to the biota and is likely to act as a temporal marker for ecological events, similar to a winter event in temperate regions. The response of the plant community to recurrent and predictable drought during the long dry season could be used to estimate the long-term response to drying over multi-annual time scales (*Detto et al., 2018*).

Reduced light availability during the long dry season in the Gabon region is most strongly associated with seasonal low-level cloud cover (*Philippon et al., 2019*). Aerosol load may also have a seasonal influence on light availability as aerosol optical depth and solar radiation appear to cycle in anti-phase although we are not able to tease apart their relative importance in this analysis (Fig. 2). Low direct solar radiation and cool temperatures will reduce water demand during these months (e.g. potential evapotranspiration is less than 2.3 mm per day during the long dry season in SW Gabon; *Philippon et al., 2019*) and are likely contributors to the forest's ability to maintain an evergreen canopy despite seasonal drought. Unsurprisingly, the savanna and forest experience different microclimates because the forest canopy creates a more humid, cooler climate throughout the year with a reduced range between daytime and night-time temperatures (Fig. 2). It is possible that the forest may also directly enhance water supply for plants during periods of low precipitation/high cloud cover due to foliar interception of low-lying clouds. At subtropical forest site (~1,000 m above sea level), foliar interception has shown to contribute an additional 40% of moisture compared to rainfall (*Hutley et al., 1997*) meaning that rain gauge data does not always accurately represent the water balance of the forest ecosystem (*Philippon et al., 2019*). While we do not have information on foliar interception of clouds at our study site (~280 m above sea level), the hydroclimatic conditions of the region do not predict occurrence of cloud-affected forest here (*Oliveira et al., 2014*). We can assume that the impact of cloud interception on water supply is negligible, although it may occur on forested hills above 600 m (e.g. the hill local to the study station known as The Camel which reaches 678 m). A dedicated research agenda would be needed to assess the any direct contribution of clouds to moisture availability, especially during the cloudy dry seasons.

We have also shown that variability in temperature and rainfall at our site is strongly influenced by global weather patterns. The most important influence on Lopé temperature is the Pacific ENSO index, with our analysis showing strong coherence between these two datasets on multi-annual scales, especially pre-2000 (Fig. 4). This result is supported by a continent-wide study showing warming throughout Africa in El Nino years

(*Collins, 2011*). None of the other oceanic indices appeared to influence Lopé temperature in a consistent way (Fig. 4). As for Lopé rainfall, the most important influence appears to be the tropical Atlantic. Rainfall cycled in phase with southern tropical Atlantic SSTs pre-2000 and in anti-phase with northern tropical Atlantic SSTs post-2000 on multi-annual scales (Fig. 4). The phase relationships between these data series indicate that higher than average rainfall at Lopé coincides with warm conditions in the south tropical Atlantic and cool conditions in the north tropical Atlantic. This result is supported by a number of other studies; *Camberlin, Janicot & Poccard (2001)* show the Atlantic dipole (cool temperatures in the north Atlantic and warm temperatures in the south tropical Atlantic) to be associated with higher than average rainfall in the region during March–May. Similarly, *Balas, Nicholson & Klotter (2007)* and *Otto et al. (2013)* demonstrate how warm conditions in the southern equatorial Atlantic (especially the Benguela coast) coincide with enhanced rainfall in the region during the dry seasons. ENSO also appears to have some influence on Lopé rainfall although the relationship is patchy (Fig. 4). The anti-phase relationship during periods of strong coherence in our analysis indicates that rainfall decreases at Lopé during El Nino events. A similar result was found among the major studies summarised in Table 1. Finally, we found little evidence of the influence of the Indian Ocean on Lopé rainfall despite published data showing reduced rainfall in western equatorial Africa coinciding with positive IOD modes (*Dezfuli & Nicholson, 2013*; *Nicholson & Dezfuli, 2013*; *Otto et al., 2013*).

Model projections of future rainfall in western equatorial Africa cover a broad spectrum and as a result, averaged model trends are close to zero. However, those models that predict drying in the region incorporate a northward shift of the rainbelt, related to cool conditions in the Gulf of Guinea (the Atlantic Cold Tongue; *James, Washington & Rowell, 2013*; Fig. 1). The strong coherence between Lopé rainfall and SSTs of the southern equatorial Atlantic (0°–20°S) at multi-annual scales in our study provides some support for the mechanisms behind these 'dry' models. Indeed, Atlantic SSTs and circulation patterns have been an important influence on Congo Basin precipitation for the past 20,000 years (*Schefuss, Schouten & Schneider, 2005*). Overall, our work supports the idea that the drivers of rainfall variability in western equatorial Africa are highly complex, with strong local and seasonal forcing from the major oceans. Land topography (e.g. the highlands of Gabon, Cameroon and eastern Africa) is also likely to be a major influence on highly localised expressions of rainfall and rainfall variability in the region (*Balas, Nicholson & Klotter, 2007*; *Dezfuli, Zaitchik & Gnanadesikan, 2015*).

### Data quality and availability

One of the major issues with climate analyses in Central Africa is the already limited and declining amount of publicly available data from weather stations in the region: The nearest weather stations to Lopé listed on the Global Historical Climatology Network (GCHN) Daily Database (*Menne et al., 2012*) are between 136 and 185 km away and there are no public data available since 1980. The World Meteorological Organisation has a minimum recommended density of weather stations eight times higher than the modern density of weather stations in Africa (*Collins, 2011*). This lack of data has a direct impact

on the quality of gridded climate data products (*Suggitt et al., 2017*) and leads to an inability to calculate daily climatic indices for the extremes (*Niang et al., 2014*). Gabon is also one of the cloudiest places on earth (http://www.acgeospatial.co.uk/the-cloudiest-place/) which leads to large uncertainties in satellite estimates, with some satellite algorithms overestimating rainfall in the region by at least a factor of two (*Balas, Nicholson & Klotter, 2007*). Finally, poor correlation between Central African rainfall and neighbouring regions, as well as variability between individual stations, suggests much local influence and further confounds the challenges of sparse data (*Balas, Nicholson & Klotter, 2007*).

The importance of maintaining long-term study sites and improving the quality and type of weather measurements in the region has been known for some time (*Clark, 2007*). However, the region is remote and there are many financial, logistical and political challenges to face when servicing field stations. One such issue is that western equatorial Africa has the highest frequency of lightning strike in the world (*Balas, Nicholson & Klotter, 2007*) leading to difficulties and great expense maintaining equipment. Lightning damage is an issue regularly confronted at Lopé and has led to major gaps in our data record. While automatic continuous measurements can provide vast amounts of detailed data relevant for ecological studies, they are also inherently more susceptible to technical failures that need expert fixes. In our experience, data gaps are more likely to go unnoticed with automatic data collection and so while we welcome new automatic methods, we recommend maintaining long-term manual records alongside for consistency.

## CONCLUSIONS

The long-term Lopé weather record has not previously been made public and is of high value in such a data poor region. Our results support regional analyses of climatic seasonality, long-term warming and the influences of the oceans on temperature and rainfall variability. However, there are some surprises; warming has occurred more rapidly than the regional products suggest and while there remains much uncertainty in the wider region, reduced rainfall over the last three decades at Lopé is in agreement with drying trends evident from less recent observational data for western equatorial Africa. The influence of the southern equatorial Atlantic (Atlantic cold tongue) on rainfall at Lopé lends support to the mechanism behind 'dry' models of future rainfall in the region.

With a climatic regime delivering on average less than 1,500 mm per year, Lopé is a globally anomalous region for evergreen tropical forest (*Reich, 1995*). Reduced water demand during the cloudy, light-deficient long dry season is likely to be the major factor facilitating persistence of evergreen forests despite seasonal drought (*Philippon et al., 2019*). It is essential that we understand the sensitivity of this seasonal cloudiness to ocean temperatures, and the viability of forest in this dry region should the clouds disappear and thus water demand increase during the seasonal drought.

We know from historic analyses that, while forests in this region have been resilient to certain levels of climatic change, they have also been susceptible to shifts back and forth between evergreen humid forests and open, fire-prone, dry forest systems and even

savannas when changes tip over certain thresholds (*Brncic et al., 2006*; *Willis et al., 2013*). The community shifts associated with drier and warmer climates have often been non-linear and dependent on ecosystem-specific resilience at local and regional scales (*Willis et al., 2013*). Carbon fertilisation and dry season cloudiness may be shielding African humid forests from the impacts of drying and warming at present. However, we urgently need reliable information on current climate and forest function and reduced uncertainties in future projections of change to inform climate change risk assessments for the western equatorial region of Central Africa.

## ACKNOWLEDGEMENTS

We acknowledge significant periods of independent data collection undertaken by Richard Parnell, Edmond Dimoto and Lee White.

### Funding

Weather data collection at SEGC, Lopé National Park, was funded by the International Centre for Medical Research in Franceville (CIRMF; 1986–2010) and by Gabon's National Parks Agency (ANPN; 2010–present) as the institutions responsible for day-to-day running of the field station during those periods. Emma Bush received funding from a Collaborative Impact Studentship co-funded by the University of Stirling and ANPN from 2013 to 2018 to prepare and analyse these data. Nils Bunnefeld received funding from the European Research Council under the European Union's H2020/ERC grant agreement no 679651 (ConFooBio). Yadvinder Malhi received funding from the UK Natural Environment Research Council (NE/1014705/1). The funders had no role in study design, data collection and analysis, decision to publish, or preparation of the manuscript.

### Grant Disclosures

The following grant information was disclosed by the authors:
International Centre for Medical Research in Franceville (CIRMF; 1986–2010).
Gabon's National Parks Agency (ANPN; 2010–present).
University of Stirling and ANPN from 2013 to 2018.
European Research Council under the European Union's: H2020/ERC: 679651.
UK Natural Environment Research Council: NE/1014705/1.

### Competing Interests

The authors declare that they have no competing interests.

### Author Contributions

- Emma R. Bush analysed the data, prepared figures and/or tables, authored or reviewed drafts of the paper, and approved the final draft.
- Kathryn Jeffery conceived and designed the experiments, performed the experiments, analysed the data, authored or reviewed drafts of the paper, and approved the final draft.

- Nils Bunnefeld analysed the data, authored or reviewed drafts of the paper, and approved the final draft.
- Caroline Tutin conceived and designed the experiments, performed the experiments, authored or reviewed drafts of the paper, and approved the final draft.
- Ruth Musgrave conceived and designed the experiments, performed the experiments, authored or reviewed drafts of the paper, and approved the final draft.
- Ghislain Moussavou performed the experiments, authored or reviewed drafts of the paper, and approved the final draft.
- Vianet Mihindou conceived and designed the experiments, performed the experiments, authored or reviewed drafts of the paper, and approved the final draft.
- Yadvinder Malhi conceived and designed the experiments, authored or reviewed drafts of the paper, and approved the final draft.
- David Lehmann performed the experiments, authored or reviewed drafts of the paper, and approved the final draft.
- Josué Edzang Ndong performed the experiments, authored or reviewed drafts of the paper, and approved the final draft.
- Loïc Makaga performed the experiments, authored or reviewed drafts of the paper, and approved the final draft.
- Katharine Abernethy conceived and designed the experiments, performed the experiments, analysed the data, authored or reviewed drafts of the paper, and approved the final draft.

## Data Availability

The 'Lopé Weather Dataset' is available at The University of Stirling's DataSTORRE (http://hdl.handle.net/11667/133).

R code to accompany all analyses is made available in Code S1.

## Supplemental Information

Supplemental information for this article can be found online at http://dx.doi.org/10.7717/peerj.8732#supplemental-information.

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
