# Peer review of "Rare ground data confirm significant warming and drying in western equatorial Africa"

_PeerJ, doi:10.7717/peerj.8732_

## Round 0.1 · original submission · Major Revisions

I have received two reviews and both of them are positive in general, which I also concur. Therefore, I invite you to submit a revision after complying with their comments.

Reviewer 1 ·

Basic reporting

Good

Experimental design

Good

Validity of the findings

Good

Additional comments

This manuscript presents an impressive 34-year ground data of temperature and rainfall and short-term data of several climatic variables from Lopé National Park, Gabon, western equatorial Africa. The seasonal and inter-annual variation, long-term trends in local weather patterns were evaluated. The authors also investigated the effects of several natural climatic oscillations on the local weather patterns. Lopé weather has a low-light, cool, and dry season. There is a warming and drying trend in Lopé. Sea surface temperatures (SSTs) of the major oceans, like ENSO and NAO, had significant effects on the inter-annual variation in Lopé

I have two primary suggestions.

First, the authors used generalized linear mixed models (GLMM) and linear mixed models (LMM) to evaluate the ocean influences on the local weather in Gabon. However, the association between the local weather and SSTs of major oceans may be transient. The relationships between these two systems may change through time, occurring at different time and periodicity. Moreover, the time for the local system to response to the regional influences (the time lags between two systems) may also change through time. Thus, it would be better if wavelet coherence analysis can be applied in this study. Wavelet coherence analysis can deal with the problems mentioned above.

Second, there are very strong seasonal cycles within the climatic time series analyzed, and these seasonal cycles (either annual or biannual) may mask the long-term trends within the time series. Please consider to remove the seasonal components within the time series before testing the relationships between local and regional systems.

Some minor suggestions follow:

It would be easier for readers to interpret the results if the authors report the delta AIC (the differences of AIC) between the best model and others. The delta AIC for the best model will be zero, which will help readers identify which model is the best.

L402-403: I would suggest to use the words like ‘the signal of biannual cycle decreased’ instead of ‘lose power’.

Figure 3F: Part of the curve for the annual cycle is masked by the figure legend. Please move the figure legend to the top left of the figure.

·

Basic reporting

The manuscript is clear, well written and easy to read. The introduction is clear and the inclusion of review of published literature on drivers of weather variability is unique and adds much value to the manuscript. However, there is the need to provide more information on the other climate variables (L33-34; absolute humidity, wind speed, solar radiation and aerosol optical depth), which was introduced in the abstract. Moreover, on L156-158 there appears to be a missing link - what may happen to these forests if warming climate push the region towards drought-adapted deciduous ecosystem? I think this portion of the introduction is incomplete. The authors have to provide the reader with potential impact/influence on the region if it dominated by drought-adapted ecosystems. Nevertheless, the figures and tables are clear and relevant (but see general comments) and the raw data have been supplied.

Experimental design

The authors have defined the research questions and set them appropriately within the context of the study. They have conducted vigorous analysis, despite the challenges with the data. However, they can improve the analysis. First, I think it will be easier to understand the GLM models used if they are represented as equations (e.g. L326-339). Further, testing whether in the different interaction terms improved the model, the authors may want to consider using a likelihood ratio test rather than depending on only the AICs to select their models (e.g. L326-339). See example in Wang et al. (2011). Second, on L175-210 the authors elaborated on the challenges they faced in collecting the long-term climate data – a problem common across Africa. Wouldn't the standardisation approach they undertook (L211-213) also introduce other biases, which may affect their findings? I also suggest the authors explain to the reader why they used a 10-day running mean of the calender year (L271-272 and others) when describing seasonality and the other trends? There is also the need for more details on how the spectral analysis was done (L278-279)

Validity of the findings

The authors have provided interpretation for the results but this should be improved. For instance, the authors analyzed seasonal trends of humidity, wind speed, solar radiation and aerosol optical depth (L34, L240-246, L365-378, Fig. 2B, Fig. 2E, Fig. 2F and Fig. 2G and 2H; Supplementary 3.1-6.1) but there was no discussion on these trends. I suggest the authors either remove the details on these variables from the entire text or provide interpretation on the seasonal patterns they observed in these climate variables. This is important because Central African forests may be susceptible/vulnerable to climate variables such as changes in solar radiation (Philippon et al. 2019). Importantly, since the data had many gaps and the fact that they only focused on minimum temperature I suggest the authors do not overstate their results in the discussion (e.g. (L508-511 'As.....in the region (as in Collins 2011’ and also L460-462). Also the link between the warming and drying observed by the authors and how they impact the survival of forests (as stated in the title) in the region was weak.

Additional comments

The authors have worked hard to get this long-term weather records in Lopé, which is notoriously difficult to obtain in most tropical African regions. This manuscript is relevant and will provide insights into the long-term climate trends and their drivers in tropical Africa. But there are other few minor issues that the authors have to consider to improve

L158: Provide reference
L454: Is the change (0.15°C) increasing or decreasing?
L479-485: Remove/delete sentence. It is a repetition of the results
L501: Remove one 'in'
L525 and 526: consistency in writing et al.
Please explain what the asterisk (*) mean (Table 4, Table 6, Table 7)

Reference
Wang et al. (2011). Modelling growth responses of individual trees to early-age thinning Eucalytus globulus, E. nitens and E. grandis plantations in northern Victoria. Australian Forestry 74: 62-72.

Philippon et al. (2019). The light-deficient climates of western Central African evergreen forests. Environmental Research Letters 14: 034007.

---

## Round 0.2 · Major Revisions

I returned your ms to the original 2 reviewers but failed to obtain feedback from both reviewers. I felt that it was necessary to have more opinions before making the final decision. Therefore, a third reviewer was invited. These comments are pretty constructive. Please incorporate those in your revision. Thank you!

Reviewer 1 ·

Basic reporting

Good

Experimental design

Good

Validity of the findings

Good

Additional comments

This manuscript presents an impressive 34-year ground data of temperature and rainfall and short-term data of several climatic variables from Lopé National Park, Gabon, western equatorial Africa. Their results indicate that there is a warming and drying trends in Lopé, and the inter-annual variations in minimum temperature and rainfall were associated with ENSO and NAO. The manuscript is well-prepared and has greatly improved from previous version. However, there are some points to follow up.

1. The authors applied wavelet coherency analysis to evaluate the associations between sea surface temperatures (SSTs) and the climatic variables in Lopé. However, they did not remove the seasonal component from the time series before the wavelet analysis. There is a strong signal of seasonality (both annual and bi-annual) in SSTs and the local climatic variables, as we can see in the Figures 3 and S3. Since this seasonality comes from other external driving forces, such the movement of ITCZ zone, and may mask other signals (inter-annual variation and long-term trends). The authors may try using additive monthly factors (subtract the long-term mean of each month from the time series) or Seasonal Decomposition of Time Series by Loess (function ‘stl’ in R) to remove the seasonal component before performing the analysis of trend detecting and wavelet coherence.

2. Figure S3 is one of the main results, while Figure 4 did not provide much information. Please consider add Figure S3 to the main text and move Figure 4 to the appendix.

3. In Fig. 3E and 3F, there are no labels indicated which one is the wavelet power of rainfall time series and which one is temperature. There is no explanation in the figure caption, either. Please add the figure labels to clarify.

Reviewer 3 ·

Basic reporting

No comment

Experimental design

1. The approach of using models to fill up the gaps of missing data may influence results of trend analyses. Therefore, please provide information about how bad the missing data problem (e.g. how many weeks of data are missing in that 34 years) is and elucidate possible impacts of this approach on your results.

Validity of the findings

No comment

Additional comments

General comments

1. It is better to remove “at a critical level for forest survival” from the title. To keep these words in the title, you need to show that tree mortality caused by drought stress is increasing in the last 3 decades. Please still keep sentences for forest survival in the discussion.

2. Annual rainfall is not a good index to quantify drought stress for plants. Plants are under drought stress when water input (precipitation) is lower than water removal (evapotranspiration). Moreover, annual rainfall contributes partially to the total water input of an area with low cloud (see the paragraph below for precipitation contributed by intercepted cloud). In other words, low annual rainfall does not lead to high tree mortality resulted from drought stress if there are additional sources of precipitation and evapotranspiration is low. Compared with rainfall, the difference between actual evapotranspiration and potential evapotranspiration and the difference between the precipitation and potential evapotranspiration are much better ways to quantify drought stress for plants (e.g. Rind et al. 1990).

3. It is better to replace “western equatorial Africa” with a small-scale location name. Balas et al. (2007) (in L604-606 of the present study) reported that rainfall varies with sites. If you want to keep “western equatorial Africa” in the title, you need to provide evidence that the trends derived from the weather data of Lopé National Park happen ubiquitously throughout western equatorial Africa.

4. Please remove the discussions for low plant photosynthesis rates resulted from light deficit and add discussions for low evapotranspiration associated with low direct solar radiation. The main finding of the present study is the warming and drying trends in your study site. Plant survival is closely related to evapotranspiration in dry seasons (i.e. water input is low). Therefore, the later discussions should be added to infer how cloudiness interacts with drying and in turn influences forest survival. Removing the former discussions will allow readers to focus on the later discussions, otherwise they are kind of distracting.

5. Please provide information about precipitation resulted from cloud intercepted by tree foliage. In addition to the reduction of evapotranspiration, low cloud can provide an importance source of precipitation for forests. For example, in a subtropical rainforest where annual rainfall measured with conventional rain gauges is as low as 1125 mm, additional precipitation resulted from cloud intercepted by tree foliage can provide up to 40% of annual rainfall (Hutley et al. 1996).

6. There is hardly low-lying cloud when it is windy. The sentence in L138-139 mentions the presence of low-lying cloud in the long dry season and the sentence in L142-143 mentions that the long dry season is the windiest season. Please explain how these contradicted situations can happen in the same place and the same period of time.

7. Please use tables to show some of your contents. For example, compared with texts in L230-260, a table with a few columns (e.g. time period, instrument manufacturer) is easier for readers to figure out how and what weather data are collected in that 34 years.

8. Please use figures and tables of the present study to support your conclusions in the discussion. For example, the sentences in L518-519, L526-528 and L563-564.

9. Please make sure that all cited references are listed in the references section. For example, Wu et al. (2016) (cited in L553) is not listed.



Specific comments

1. Please mark the location of the weather station in Figure 1.
2. Please insert space between values and their units. For example, “2000mm” in L93 should be “2000 mm” and “800m” in L229 should be “800 m”.
3. L532, L534, L536: “°c” should be “°C”.
4. L535-536: Add “+” before “0.26” and “0.22”.
5. L535-536: Is the warming rate (+0.26°C) of America correct? The warming rate (+0.25°C) of your study site is actually lower than +0.26°C.
6. Figure number is missing. For example, “Fig. 2C and D” in L441 should be “Fig. 2C and 2D”.
7. L632: “Iit” should be “It”.



Reference
1. Rind, D., Goldberg, R., Hansen, J., Rosenzweig, C., & Ruedy, R. (1990). Potential evapotranspiration and the likelihood of future drought. Journal of Geophysical Research: Atmospheres, 95(D7), 9983-10004.
2. Hutley, L. B., Doley, D., Yates, D. J., & Boonsaner, A. (1997). Water balance of an Australian subtropical rainforest at altitude: the ecological and physiological significance of intercepted cloud and fog. Australian Journal of Botany, 45(2), 311-329.

---

## Round 0.3 · accepted · Accept

I have received comments from the 3rd reviewer; all of those are cosmetic. Therefore, I am pleased to inform you that your paper "Rare ground data confirm significant warming and drying in western equatorial Africa" has been accepted for publication in PeerJ. Super congrats!

Reviewer 3 ·

Basic reporting

1. Please always place a space between the numerical value and unit symbol. For example, in L564, “2.3mm” should be ”2.3 mm”.
proper: 2 m, 15 mm
improper: 2m, 15mm

For more information, please refer to the websites below.
https://www.monash.edu/about/editorialstyle/editing/numbers
https://physics.nist.gov/cuu/Units/checklist.html

2. L571: It is a subtropical rather than tropical forest.

3. L492: “+0.19Cc”?

4. Reference formats are not uniform. I listed three inconsistencies as examples. Please systematically double check the formats of all references and make sure they are uniform.
i. Some references show “pp.” (L706, L717, L751, L760, L792……) but most references don’t.
ii. Page number is missing in L863.
iii. Please remove underscores and asterisks from L947-948.

Experimental design

NA

Validity of the findings

NA